# MODELING TEMPORAL DATA AS CONTINUOUS FUNCTIONS WITH PROCESS DIFFUSION

## ABSTRACT

Temporal data like time series are often observed at irregular intervals which is a challenging setting for existing machine learning methods. To tackle this problem, we view such data as samples from some underlying continuous function. We then define a diffusion-based generative model that adds noise from a predefined stochastic process while preserving the continuity of the resulting underlying function. A neural network is trained to reverse this process which allows us to sample new realizations from the learned distribution. We define suitable stochastic processes as noise sources and introduce novel denoising and score-matching models on processes. Further, we show how to apply this approach to the multivariate probabilistic forecasting and imputation tasks. Through our extensive experiments, we demonstrate that our method outperforms previous models on synthetic and real-world datasets.

## 1 INTRODUCTION

Time series data is collected from measurements of some real-world system that evolves via some complex unknown dynamics and the sampling rate is often arbitrary and non-constant. Thus, the assumption that time series follows some underlying continuous function is reasonable; consider, e.g., the temperature or load of a system over time. Although the values are observed as separate events, we know the temperature always exists and its evolution over time is smooth, not jittery. The continuity assumption remains when the intervals between the measurements vary. This kind of data can be found in many domains, from medical, industrial to financial applications.

Different approaches to model irregular data have been proposed, including neural (ordinary or stochastic) differential equations (Chen et al., 2018; Li et al., 2020), neural processes (Garnelo et al., 2018), normalizing flows (Deng et al., 2020) etc. As it turns out, capturing the true generative process proves to be difficult, especially with the inherent stochasticity of the data.

We propose an alternative method, a generative model for continuous data that is based on the diffusion framework (Ho et al., 2020) which simply adds noise to a data point until it contains no information about the original input. At the same time, the generative part of these models learns to reverse this process so that we can sample new realizations once training is completed. In this paper, we apply these ideas to the time series setting and address the unique challenges that arise.

**Contributions.** Contrary to the previous works on diffusion, we model continuous *functions*, not vectors (Fig. 1). To do so, we first define a suitable noising process that will preserve continuity. Next, we derive the transition probabilities to perform the noising and specify the evidence bound on the likelihood as well as the new sampling procedure. Finally, we propose new models that take in the noisy input and produce the denoised output or, alternatively, the value of the score function.

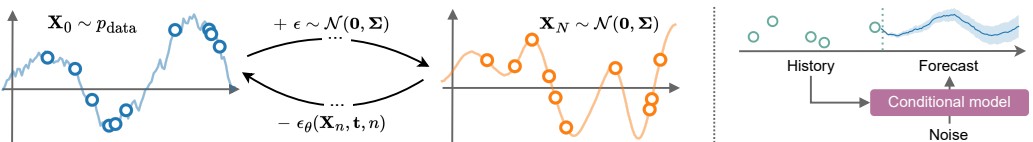

Figure 1: (Left) We add noise from a stochastic process to the *whole* time series at once. The model $\epsilon_\theta$ learns to reverse this process. (Right) We can use this approach to, e.g., forecast with uncertainty.

## 2 BACKGROUND

Given training data $\{\boldsymbol{x}\}$ where each $\boldsymbol{x} \in \mathbb{R}^d$, the goal of generative modeling is to learn the probability density function $p(\boldsymbol{x})$ and be able to generate new samples from this learned distribution. Diffusion models (Ho et al., 2020; Song et al., 2021) achieve both of these goals by learning to reverse some fixed process that adds noise to the data. In the following, we present a brief overview of the two ways to define diffusion; in Section 2.1 the noise is added across $N$ increasing scales, which is then taken to the limit in Section 2.2 by defining the diffusion using a stochastic differential equation (SDE).

### 2.1 FIXED-STEP DIFFUSION

Sohl-Dickstein et al. (2015); Ho et al. (2020) propose the denoising diffusion probabilistic model (DDPM) which gradually adds *fixed* Gaussian noise to the observed data point $\boldsymbol{x}_0$ via known scales $\beta_n$ to define a sequence of progressively noisier values $\boldsymbol{x}_1, \boldsymbol{x}_2, \ldots, \boldsymbol{x}_N$. The final noisy output $\boldsymbol{x}_N \sim \mathcal{N}(\boldsymbol{0}, \boldsymbol{I})$ carries no information about the original data point and thus the sequence of positive noise (variance) scales $\beta_1, \ldots, \beta_N$ has to be increasing such that the first noisy output $\boldsymbol{x}_1$ is close to the original data $\boldsymbol{x}_0$, and the final value $\boldsymbol{x}_N$ is pure noise. The goal is then to learn to reverse this process.

As diffusion forms a Markov chain, the transition between any two consecutive points is defined with a conditional probability $q(\boldsymbol{x}_n|\boldsymbol{x}_{n-1}) = \mathcal{N}(\sqrt{1-\beta_n}\boldsymbol{x}_{n-1}, \beta_n\boldsymbol{I})$. Since the transition kernel is Gaussian, the value at any step $n$ can be sampled directly from $\boldsymbol{x}_0$. Given $\alpha_n = 1 - \beta_n$ and $\bar{\alpha}_n = \prod_{k=1}^n \alpha_k$, we can write:

$$q(\boldsymbol{x}_n|\boldsymbol{x}_0) = \mathcal{N}(\sqrt{\bar{\alpha}_n}\boldsymbol{x}_0, (1-\bar{\alpha}_n)\boldsymbol{I}). \tag{1}$$

Further, the probability of any intermediate value $\boldsymbol{x}_{n-1}$ given its successor $\boldsymbol{x}_n$ and initial $\boldsymbol{x}_0$ is

$$q(\boldsymbol{x}_{n-1}|\boldsymbol{x}_n, \boldsymbol{x}_0) = \mathcal{N}(\tilde{\boldsymbol{\mu}}_n, \tilde{\beta}_n\boldsymbol{I}), \tag{2}$$

$$\text{where:} \quad \tilde{\boldsymbol{\mu}}_n = \frac{\sqrt{\bar{\alpha}_{n-1}}\beta_n}{1-\bar{\alpha}_n}\boldsymbol{x}_0 + \frac{\sqrt{\alpha_n}(1-\bar{\alpha}_{n-1})}{1-\bar{\alpha}_n}\boldsymbol{x}_n, \qquad \tilde{\beta}_n = \frac{1-\bar{\alpha}_{n-1}}{1-\bar{\alpha}_n}\beta_n. \tag{3}$$

The generative model learns the reverse process $p(\boldsymbol{x}_{n-1}|\boldsymbol{x}_n)$. Sohl-Dickstein et al. (2015) set $p(\boldsymbol{x}_{n-1}|\boldsymbol{x}_n) = \mathcal{N}(\boldsymbol{\mu}_\theta(\boldsymbol{x}_n, n), \beta_n\boldsymbol{I})$, and parameterized $\boldsymbol{\mu}_\theta$ with a neural network. The training objective is to maximize the evidence lower bound $\log p(\boldsymbol{x}_0) \geq$

$$\mathbb{E}_q\left[D_{\mathrm{KL}}(q(\boldsymbol{x}_N|\boldsymbol{x}_0)||p(\boldsymbol{x}_N)) + \sum_{n>1} D_{\mathrm{KL}}(q(\boldsymbol{x}_{n-1}|\boldsymbol{x}_n, \boldsymbol{x}_0)||p(\boldsymbol{x}_{n-1}|\boldsymbol{x}_n)) - \log p(\boldsymbol{x}_0|\boldsymbol{x}_1)\right]. \tag{4}$$

In practice, however, the approach by Ho et al. (2020) is to reparameterize $\boldsymbol{\mu}_\theta$ and predict the noise $\boldsymbol{\epsilon}$ that was added to $\boldsymbol{x}_0$, using a neural network $\boldsymbol{\epsilon}_\theta(\boldsymbol{x}_n, n)$, and minimize the simplified loss function:

$$\mathcal{L}(\boldsymbol{x}_0) = \mathbb{E}_{\boldsymbol{\epsilon}\sim\mathcal{N}(\boldsymbol{0},\boldsymbol{I}), n\sim\mathcal{U}(\{0,\ldots,N\})}\left[||\boldsymbol{\epsilon}_\theta(\sqrt{\bar{\alpha}_n}\boldsymbol{x}_0 + \sqrt{1-\bar{\alpha}_n}\boldsymbol{\epsilon}, n) - \boldsymbol{\epsilon}||_2^2\right]. \tag{5}$$

To generate new data, the first step is to sample a point from the final distribution $\boldsymbol{x}_N \sim \mathcal{N}(\boldsymbol{0}, \boldsymbol{I})$ and then iteratively denoise it using the above model ($\boldsymbol{x}_N \mapsto \boldsymbol{x}_{N-1} \mapsto \cdots \mapsto \boldsymbol{x}_0$) to get a sample from the data distribution. To summarize, the forward process adds the noise $\boldsymbol{\epsilon}$ to the input $\boldsymbol{x}_0$, at different scales, to produce $\boldsymbol{x}_n$. The model learns to invert this, i.e., predicts the noise $\boldsymbol{\epsilon}$ from $\boldsymbol{x}_n$.

### 2.2 SDE DIFFUSION

Instead of taking a finite number of diffusion steps as in Section 2.1, Song et al. (2021) introduce a continuous diffusion of vector valued data, $\boldsymbol{x}_0 \mapsto \boldsymbol{x}_s$ where $s \in [0, S]$ is now a continuous variable. The forward process can be elegantly defined with an SDE:

$$\mathrm{d}\boldsymbol{x}_s = f(\boldsymbol{x}_s, s)\mathrm{d}s + g(s)\mathrm{d}W_s, \tag{6}$$

where $W$ is a standard Wiener process. The variable $s$ is the continuous analogue of the discrete steps implying that the input gets noisier during the SDE evolution. The final value $\boldsymbol{x}_S \sim p(\boldsymbol{x}_S)$ will follow some predefined distribution, as in Section 2.1. For the forward SDE in Equation 6 there exist a corresponding reverse SDE (Anderson, 1982):

$$\mathrm{d}\boldsymbol{x}_s = [f(\boldsymbol{x}_s, s) - g(s)^2 \nabla_{\boldsymbol{x}_s} \log p(\boldsymbol{x}_s)]\mathrm{d}s + g(s)\mathrm{d}W_s, \tag{7}$$

where $\nabla_{\boldsymbol{x}_s} \log p(\boldsymbol{x}_s)$ is the score function. Solving the above SDE from $S$ to $0$, given initial condition $\boldsymbol{x}_S \sim p(\boldsymbol{x}_S)$ returns a sample from the data distribution. The generative model's goal is to learn the score function via a neural network $\psi_{\boldsymbol{\theta}}(\boldsymbol{x}_s, s)$, by minimizing the following loss:

$$\mathcal{L}(\boldsymbol{x}_0) = \mathbb{E}_{\boldsymbol{x}_s \sim \text{SDE}(\boldsymbol{x}_0), s \sim \mathcal{U}(0,S)} \left[ ||\psi_{\boldsymbol{\theta}}(\boldsymbol{x}_s, s) - \nabla_{\boldsymbol{x}_s} \log p(\boldsymbol{x}_s)||_2^2 \right]. \tag{8}$$

Song et al. (2021) define the continuous equivalent to DDPM forward process as the following SDE:

$$\mathrm{d}\boldsymbol{x}_s = -\frac{1}{2}\beta(s)\boldsymbol{x}_s\mathrm{d}s + \sqrt{\beta(s)}\mathrm{d}W_s, \tag{9}$$

where $\beta(s)$ and $S$ are chosen in such a way that ensures the final noise distribution is unit normal, $\boldsymbol{x}_S \sim \mathcal{N}(\mathbf{0}, \boldsymbol{I})$. Given this specific parameterization, one can easily derive the transition probability $q(\boldsymbol{x}_s|\boldsymbol{x}_0)$ and calculate the exact score in closed-form (see Section 3.3 and Appendix A.3).

## 3 DIFFUSION FOR TIME SERIES DATA

In contrast to the previous section which deals with data points that are represented by vectors, we are interested in generative modeling for time series data. We represent time series as a set of points $\boldsymbol{X} = \{\boldsymbol{x}(t_0), \dots, \boldsymbol{x}(t_{M-1})\}$, $t \in [0, T]$, observed across $M$ timestamps. The observations can be equally spaced but this formulation encompasses irregularly-sampled data as well. We assume that each observed time series comes from its corresponding underlying continuous function $\boldsymbol{x}(\cdot)$.

Our approach can be viewed as modeling the distribution "$p(\boldsymbol{x}(\cdot))$" over functions instead of vectors, which amounts to learning the stochastic process. To preserve continuity, we cannot apply the ideas from Section 2 directly, unless we assume measurements are independent of each other. The issue of adding an independent noise in the diffusion arises because it produces discontinuous samples, which is at odds with our assumption. In the following, we address this and propose a solution.

### 3.1 STOCHASTIC PROCESSES AS NOISE SOURCES FOR DIFFUSION

Instead of defining the diffusion by adding some scaled noise vector $\boldsymbol{\epsilon} \sim \mathcal{N}(\mathbf{0}, \boldsymbol{I})$ to a data vector $\boldsymbol{x}$, we add a noise *function* (stochastic process) $\boldsymbol{\epsilon}(\cdot)$ to the underlying data function $\boldsymbol{x}(\cdot)$. The only restriction on $\boldsymbol{\epsilon}(\cdot)$ is that it has to be continuous so that the output remains continuous as well, which clearly rules out stochastic processes where time is indexed by a *finite* set e.g. $\boldsymbol{\epsilon}(\cdot) \sim \mathcal{N}(\mathbf{0}, \boldsymbol{I})$. However, using a normal distribution proved to be very convenient in Section 2 as it allows for closed-form formulations of various terms, especially the loss. This is due to the fact that adding two Gaussian random variables leads to another Gaussian variable, which, as an aside, is a property shared by some other parametric distributions (Nachmani et al., 2021).

Therefore, our goal is to define $\boldsymbol{\epsilon}(\cdot)$ which will satisfy the continuity property while giving us tractable training and sampling. Note that $t$ refers to the time of the observation and $\boldsymbol{\epsilon}(t)$ is the noise at $t$, in contrast to the previous section where *time-like* variables $n$ and $s$ referred to the noise scale.

We could consider obtaining the noise from a standard Wiener process $\boldsymbol{\epsilon}(\cdot) = W(\cdot)$. Although we will have all the terms in closed-form, a clear disadvantage of this approach is that variance grows with time. Additionally, the distribution of $W(0)$ is degenerate as we never add any noise. This can be solved in an ad hoc manner by shifting the whole time series similar to Deng et al. (2020).

Instead, in the following, we present two *stationary* stochastic processes that add the same amount of noise regardless of the time of the observation. Note that the noise is *correlated* in the time dimension, hence the use of the stochastic process. An additional nice property of these processes is that they reduce to the diffusion from Section 2 in the trivial case of time series with only one element.

**1. Gaussian process prior.** Given a set of $M$ time points $\boldsymbol{t}$, we propose sampling $\boldsymbol{\epsilon}(t)$, $t \in \boldsymbol{t}$ from a Gaussian process $\mathcal{N}(\mathbf{0}, \boldsymbol{\Sigma})$, where each element of the covariance matrix is specified with a kernel $\text{cov}(t, u) = k(t, u)$. This produces *smooth* noise functions $\boldsymbol{\epsilon}(\cdot)$ that can be evaluated at any $t$.

To define a stationary process, we have to use a stationary kernel; we will use a radial basis kernel $k(t, u) = \exp(-\gamma(t-u)^2)$. Adjusting the parameter $\gamma$ lets us vary the flatness of the noise curves. Given a set of time points $\boldsymbol{t}$, we can easily sample from this process by first computing the covariance $\boldsymbol{\Sigma} = k(\boldsymbol{t}, \boldsymbol{t})$ and then sample from the multivariate normal distribution $\mathcal{N}(\mathbf{0}, \boldsymbol{\Sigma})$.

| **Algorithm 1** Loss (DSPD-GP diffusion) | **Algorithm 2** Sampling (DSPD-GP diffusion) |
|---|---|
| 1: $\boldsymbol{X}_0, \boldsymbol{t} \sim p_{\text{data}}(\boldsymbol{X}, \boldsymbol{t})$ | 1: **input:** $\boldsymbol{t} = \{t_0, \ldots, t_{M-1}\}$ |
| 2: $\boldsymbol{\Sigma} = k(\boldsymbol{t}, \boldsymbol{t})$ | 2: $\boldsymbol{\Sigma} = k(\boldsymbol{t}, \boldsymbol{t}); \boldsymbol{L} = \text{Cholesky}(\boldsymbol{\Sigma})$ |
| 3: $\boldsymbol{L} = \text{Cholesky}(\boldsymbol{\Sigma})$ | 3: $\boldsymbol{X}_N \sim \mathcal{N}(\boldsymbol{0}, \boldsymbol{\Sigma})$ |
| 4: $\tilde{\boldsymbol{\epsilon}} \sim \mathcal{N}(\boldsymbol{0}, \boldsymbol{I})$ | 4: **for** $n = N, \ldots, 1$ **do** |
| 5: $\boldsymbol{\epsilon} = \boldsymbol{L}\tilde{\boldsymbol{\epsilon}}$ | 5: $\quad \boldsymbol{z} \sim \mathcal{N}(\boldsymbol{0}, \boldsymbol{\Sigma})$ |
| 6: $n \sim \mathcal{U}(\{1, \ldots, N\})$ | 6: $\quad \boldsymbol{X}_{n-1} = \frac{1}{\sqrt{\alpha_n}} \left( \boldsymbol{X}_n - \frac{1-\alpha_n}{\sqrt{1-\bar{\alpha}_n}} \boldsymbol{L}\boldsymbol{\epsilon}_\theta(\boldsymbol{X}_n, \boldsymbol{t}, n) \right) + \beta_n \boldsymbol{z}$ |
| 7: $\boldsymbol{X}_n = \sqrt{\bar{\alpha}_n}\boldsymbol{X}_0 + \sqrt{1 - \bar{\alpha}_n}\boldsymbol{\epsilon}$ | 7: **end for** |
| 8: $\mathcal{L} = \|\tilde{\boldsymbol{\epsilon}} - \boldsymbol{\epsilon}_\theta(\boldsymbol{X}_n, \boldsymbol{t}, n)\|_2^2$ | 8: **return** $\boldsymbol{X}_0$ |

**2. Ornstein-Uhlenbeck diffusion.** The alternative noise distribution is a stationary OU process which is specified as a solution to the following SDE:

$$\mathrm{d}\boldsymbol{\epsilon}_t = -\gamma\boldsymbol{\epsilon}_t\mathrm{d}t + \mathrm{d}W_t, \tag{10}$$

where $W_t$ is the standard Wiener process and we use the initial condition $\boldsymbol{\epsilon}_0 \sim \mathcal{N}(\boldsymbol{0}, \boldsymbol{I})$. We can obtain samples from OU process easily by sampling from a time-changed and scaled Wiener process: $e^{-\gamma t}W_{e^{2\gamma t}}$. The covariance can be calculated as $\text{cov}(t, u) = \exp(-\gamma|t - u|)$. The OU process is a special case of a Gaussian process with a Matérn kernel ($\nu = 0.5$) (Rasmussen & Williams, 2005, p. 86). We discuss different sampling techniques and their trade-offs in Appendix A.4.

In the end, both the GP and OU processes are defined with a multivariate normal distribution $\mathcal{N}(\boldsymbol{0}, \boldsymbol{\Sigma})$, where $\boldsymbol{\Sigma}$ is calculated using the times of the observations. As opposed to the diffusions from Section 2, we use correlated noise in the forward process. Our approach allows us to produce continuous functions as samples and will prove to be a natural way to do forecasting and imputation.

### 3.2 DISCRETE STOCHASTIC PROCESS DIFFUSION (DSPD)

We apply the discrete diffusion framework to the time series setting. Reusing the notation from before, $\boldsymbol{X}_0$ denotes the input data and $\boldsymbol{X}_n = \{\boldsymbol{x}_n(t_0), \ldots, \boldsymbol{x}_n(t_{M-1})\}$ is the noisy output after $n$ diffusion steps. In contrast to the classical DDPM (Section 2.1) where one adds independent Gaussian noise to data, we now add the noise from a stochastic process. In particular, given the times of the input observations, we can compute the covariance $\boldsymbol{\Sigma}$ and sample noise $\boldsymbol{\epsilon}(\cdot)$ from a GP or OU process as defined in Section 3.1. We can write the transition kernel and the posterior as:

$$q(\boldsymbol{X}_n|\boldsymbol{X}_0) = \mathcal{N}(\sqrt{\bar{\alpha}_n}\boldsymbol{X}_0, (1 - \bar{\alpha}_n)\boldsymbol{\Sigma}), \tag{11}$$

$$q(\boldsymbol{X}_{n-1}|\boldsymbol{X}_n, \boldsymbol{X}_0) = \mathcal{N}(\tilde{\boldsymbol{\mu}}_n, \tilde{\beta}_n\boldsymbol{\Sigma}), \tag{12}$$

where the difference to Equations 1 and 2 is the inclusion of $\boldsymbol{\Sigma}$. Full derivation is in Appendix A.1.

The generative model is defined with the reverse process $p(\boldsymbol{X}_{n-1}|\boldsymbol{X}_n) = \mathcal{N}(\boldsymbol{\mu}_\theta(\boldsymbol{X}_n, \boldsymbol{t}, n), \beta_n\boldsymbol{\Sigma})$, similar to Ho et al. (2020), but we swap the identity matrix for $\boldsymbol{\Sigma}$. Another key difference is that the model now takes the full time series and the time points in order to output the prediction which has the same size as $\boldsymbol{X}_n$. The architecture, therefore, has to be a type of a time series encoder-decoder.

Since all the probabilities are still normal, the terms in the ELBO (Equation 4) can be calculated in closed-form. Following Ho et al. (2020), we also adopt the reparameterization which predicts the noise $\boldsymbol{\epsilon}(\cdot)$ and use the simplified loss as in Equation 5:

$$\mathcal{L}(\boldsymbol{X}_0) = \mathbb{E}_{\boldsymbol{\epsilon}\sim\mathcal{N}(\boldsymbol{0},\boldsymbol{\Sigma}),n\sim\mathcal{U}(\{1,\ldots,N\})}\left[\|\boldsymbol{\epsilon}_\theta(\sqrt{\bar{\alpha}_n}\boldsymbol{X}_0 + \sqrt{1 - \bar{\alpha}_n}\boldsymbol{\epsilon}, \boldsymbol{t}, n) - \boldsymbol{\epsilon}\|_2^2\right]. \tag{13}$$

More details can be found in Appendix A.2. The sampling is similar to Ho et al. (2020) but the noise comes from a stochastic process instead of an independent normal distributions. In Algorithms 1 and 2 we show the training and the sampling procedure, respectively, when using the GP diffusion.

### 3.3 CONTINUOUS STOCHASTIC PROCESS DIFFUSION (CSPD)

Similarly to the previous section, we can extend the continuous diffusion framework to use the noise coming from a Gaussian or OU process. Now, the noise scales $\beta(s)$ are continuous in the diffusion

time $s$, see Section 2.2. Given a factorized covariance matrix $\boldsymbol{\Sigma} = \boldsymbol{L}\boldsymbol{L}^T$, we modify the variance preserving diffusion SDE (Song et al., 2021):

$$\mathrm{d}\boldsymbol{X}_s = -\frac{1}{2}\beta(s)\boldsymbol{X}_s\mathrm{d}s + \sqrt{\beta(s)}\boldsymbol{L}\mathrm{d}W_s, \tag{14}$$

which gives us the following transition probability (see Appendix A.3 for details):

$$q(\boldsymbol{X}_s|\boldsymbol{x}_0) = \mathcal{N}(\tilde{\boldsymbol{\mu}}, \tilde{\boldsymbol{\Sigma}}) = \mathcal{N}\left(\boldsymbol{X}_0 e^{-\frac{1}{2}\int_0^s \beta(s)\mathrm{d}s}, \boldsymbol{\Sigma}\left(1 - e^{-\int_0^s \beta(s)\mathrm{d}s}\right)\right). \tag{15}$$

Since this probability is normal, the value of the score function can be computed in closed-form:

$$\nabla_{\boldsymbol{X}_s}\log q(\boldsymbol{X}_s|\boldsymbol{X}_0) = -\tilde{\boldsymbol{\Sigma}}^{-1}(\boldsymbol{X}_s - \tilde{\boldsymbol{\mu}}), \tag{16}$$

which we can use to optimize the same objective as in Equation 8. Our neural network $\boldsymbol{\epsilon}_\theta(\boldsymbol{X}_s, \boldsymbol{t}, s)$ will take in the full time series, together with the observation times $\boldsymbol{t}$ and the diffusion time $s$, and predict the values of the score function. As it turns out, we can again use the reparameterization in which we predict the noise, whilst the score is only calculated when sampling new realizations.

### 3.4 IMPLEMENTATION DETAILS

In our work, we consider multivariate time series which means each observation at a certain time point is a $d$-dimensional vector. In the forward diffusion process we treat the data as $d$ individual univariate time series and add the noise to them independently. This is in line with the previous works where, e.g., independent noise is added to the individual pixels in an image. The model takes in a complete noisy multivariate time series to learn the reverse process so it handles the correlations between the data dimensions and across the time dimension.

We note that the best results are obtained if the model is reparameterized to always predict the independent Gaussian noise. In the discrete stochastic process diffusion, this means the noise is computed as $\boldsymbol{\epsilon} = \boldsymbol{L}\tilde{\boldsymbol{\epsilon}}, \tilde{\boldsymbol{\epsilon}} \sim \mathcal{N}(\boldsymbol{0}, \boldsymbol{I})$, where $\boldsymbol{L}^T\boldsymbol{L} = \boldsymbol{\Sigma}$ is the covariance matrix of the stochastic process. The model then learns to predict $\tilde{\boldsymbol{\epsilon}}$. Similarly, in the continuous diffusion, we represent the score as $\boldsymbol{L}\tilde{\boldsymbol{\epsilon}}/\sigma^2$, where $\sigma^2 = 1 - \exp(-\int_0^s \beta(s)\mathrm{d}s)$ (Equation 16). In both cases, when we sample, the model will output the prediction of $\tilde{\boldsymbol{\epsilon}}$, which we transform to get the sample from the stochastic process or to obtain the score function value. An example of this can be found in Algorithm 2 for the discrete diffusion.

## 4 APPLICATIONS

To train a generative model, it must learn to reverse the forward diffusion process by predicting the noise that was added to the clean data. The input to the model is the time series $(\boldsymbol{X}_0, \boldsymbol{t})$ along with the diffusion step $n$ or diffusion time $s$, and the output is of the same size as $\boldsymbol{X}_0$. If additional inputs are available, we can also model the conditional distribution; for example, time series data often contains covariates for each time point of $\boldsymbol{t}$. We can also condition the generation on the past observations which essentially defines a probabilistic forecaster or condition only on the observed values which defines a neural process or an imputation model.

### 4.1 FORECASTING MULTIVARIATE TIME SERIES

Forecasting is answering what is going to happen, given what we have seen, and as such is the most prominent tasks in time series analysis. Probabilistic forecasting adds the layer of (aleatoric) uncertainty on top of that and returns the confidence intervals which is often a requirement for deploying models in real world settings. The neural forecasters are usually encoder-decoders, where the history of observations $(\boldsymbol{X}^H, \boldsymbol{t}^H)$ is represented with a single vector $\boldsymbol{z}$ and the decoder outputs the distribution of the future values $\boldsymbol{X}^F$ given $\boldsymbol{z}$ at time points $\boldsymbol{t}^F$. Previous works relied on specifying the parameters of the output distribution, e.g., via a diagonal covariance (Salinas et al., 2020) or some low-rank approximation (Salinas et al., 2019), relying on normalizing flows (de Bézenac et al., 2020; Rasul et al., 2021b), or Generative Adversarial Networks (GANs) (Koochali et al., 2021).

Recently, Rasul et al. (2021a) introduced a diffusion based forecasting model to learn the conditional probability $p(\boldsymbol{X}^F|\boldsymbol{X}^H)$. In particular, let $\boldsymbol{X}^H = \{\boldsymbol{x}(t_0), \ldots, \boldsymbol{x}(t_{M-1})\}$ be a history window of

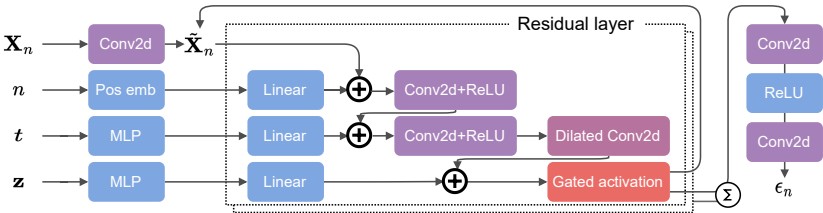

Figure 2: Overview of the forecasting model. The inputs are time series $\boldsymbol{X}_n$, diffusion step $n$, observation times $\boldsymbol{t}$ and the history vector $\boldsymbol{z}$. The output is the predicted noise value $\boldsymbol{\epsilon}_n$.

size $M$ sampled randomly from the full training data. They specify the distribution $p(\boldsymbol{x}(t_M)|\boldsymbol{X}^H)$ using a conditional DDPM model. The forward process adds independent Gaussian noise to $\boldsymbol{x}(t_M)$ the same way as in DDPM. However, the reverse denoising model is conditioned on the history $\boldsymbol{X}^H$ which is represented with a fixed sized vector $\boldsymbol{z}$. After training is completed, the predictions are made in the following way: (1) $\boldsymbol{X}^H$ is encoded with an RNN to obtain $\boldsymbol{z}$; (2) the initial noisy value is sampled $\boldsymbol{x}_N(t_M) \sim \mathcal{N}(\boldsymbol{0}, \boldsymbol{I})$; and (3) denoising is performed using the sampling algorithm from Ho et al. (2020) but conditioned on $\boldsymbol{z}$ to obtain $\boldsymbol{x}_0(t_M)$. The final denoised value is the forecast and sampling multiple values allows computing empirical confidence intervals of interest.

In Rasul et al. (2021a), the timestamps are always discrete and the prediction is autoregressive, i.e., the values are produced *one by one*. Our diffusion framework offers two key improvements: first, the predictions can be made at any future time point (in continuous time); and second, we can predict multiple values at the same time which scales better on modern hardware.

In our case, the prediction $\boldsymbol{X}^F$ will not be a single vector but a set $\{\boldsymbol{x}(t_M), \ldots, \boldsymbol{x}(t_{M+K})\}$ of size $K$. This type of data is naturally handled by process diffusion as defined in Section 3. The only thing left to do is to design a suitable denoising model $\boldsymbol{\epsilon}_\theta$. Previous observations are again represented with an RNN to obtain $\boldsymbol{z}$ and condition the reverse process on it. We propose an architecture similar to the TimeGrad model from Rasul et al. (2021a) but which outputs multiple values simultaneously. Figure 2 shows the architecture overview: we add $\boldsymbol{t}$ as an input and take the whole time series at once. Thus, we use 2D convolution where the extra dimension corresponds to the time dimension.

## 4.2 DIFFUSION PROCESS AS A NEURAL PROCESS

Neural processes (Garnelo et al., 2018) are a class of latent variable models that define a stochastic process with neural networks. Given a set of data points (a dataset), the model outputs the probability distribution over the functions that generated this dataset. That is, for different datasets, the model will define different stochastic processes. Due to this behavior, neural processes bear a resemblance to the Gaussian processes but can also be viewed as a meta learning model (Hospedales et al., 2021).

Let $\boldsymbol{X}^A$ denote the observed data, in our case, a time series, and let $\boldsymbol{X}^B$ be the unobserved data at the time points $\boldsymbol{t}^B$. Garnelo et al. (2018) construct the encoder-decoder model that uses the amortized variational inference for training (Kingma & Welling, 2014). The encoder takes in a set of observed points $(\boldsymbol{X}^A, \boldsymbol{t}^A)$ and outputs the distribution over the latent variable $q(\boldsymbol{z})$. It is crucial that the encoder is permutation invariant, i.e., the order in of the input points does not alter the result. This is easy to achieve using, e.g., deep sets (Zaheer et al., 2017). The decoder takes in the sampled latent vector $\boldsymbol{z}$ and the query time points $\boldsymbol{t}^B$ and predicts the values of the unobserved points $\boldsymbol{X}^B$.

Since our approach samples functions, we can condition the generation on an input dataset $(\boldsymbol{X}^A, \boldsymbol{t}^A)$ in order to create our version of a neural process, based purely on the diffusion framework. The encoder will be a deterministic neural network that outputs the latent vector $\boldsymbol{z}$, contrary to (Garnelo et al., 2018) which outputs the distribution. Similar to Section 4.1, the diffusion is conditioned on $\boldsymbol{z}$ and we can output samples for any query $\boldsymbol{t}^B$. Therefore, we capture the distribution $p(\boldsymbol{X}^B|\boldsymbol{X}^A)$ directly. During training, we adopt the approach of feeding in the data such that we learn $p(\boldsymbol{X}^A \cup \boldsymbol{X}^B|\boldsymbol{X}^A)$ which helps the model learn to output high certainty around $\boldsymbol{t}^A$, see Garnelo et al. (2018).

In the end, our model sees many observed-unobserved pairs coming from different true underlying processes. The model learns to represent the observed points $\boldsymbol{X}^A$ such that the denoising process

corresponds to the correct distribution, given $\boldsymbol{X}^A$. After training is completed, we take a time series $\boldsymbol{X}^A$ and output the samples at any set of query time points $\boldsymbol{t}^B$. We can view such an approach as an interpolation or imputation model that fills-in the missing values across time. The main appeal is the ability to capture different stochastic processes within a single model.

### 4.3 PROBABILISTIC TIME SERIES IMPUTATION

Previous section considered interpolating points in time. Now, we look into filling-in the missing values across the observation dimensions, i.e., the imputation of the vectors. An element $\boldsymbol{x}$ of the time series $\boldsymbol{X}$ is assigned a mask $\boldsymbol{m}$ of the same dimension that indicates whether the $i$-th value $x_i$ has been observed ($m_i = 1$) or is missing ($m_i = 0$).

Given observed $\boldsymbol{X}^A$ and missing points $\boldsymbol{X}^B$, Tashiro et al. (2021) propose a model that learns a conditional distribution $p(\boldsymbol{X}^B|\boldsymbol{X}^A)$. The model is built upon a diffusion framework and the reverse process is conditioned on $\boldsymbol{X}^A$, similar to Section 4.2. We extend this by introducing noise from a stochastic process, as presented above. The learnable model remains the same but we introduce the correlated noise in the loss and sampling. We posit that continuous noise process as an inductive bias for the irregular time series is a more natural choice.

## 5 RELATED WORK

Generative modeling with diffusion (Sohl-Dickstein et al., 2015; Ho et al., 2020; Song et al., 2021) recently gained traction as it provides good sampling quality on image generation (Dhariwal & Nichol, 2021; Ramesh et al., 2022; Rombach et al., 2021) and became the state-of-the-art method replacing GANs (Goodfellow et al., 2020). The modeling power translates to other tasks as well, so it has been used in modeling, e.g., waveforms (Kong et al., 2021) and time series forecasting (Rasul et al., 2021a), but also discrete data such as text (Austin et al., 2021) and molecules (Anand & Achim, 2022; Lee & Kim, 2022).

The number of steps can be either fixed (Ho et al., 2020) or the steps can be continuously indexed (Song et al., 2021; Huang et al., 2021). Based on this, the reverse process is usually defined as denoising or score-based, respectively. We implement both of the approaches in our stochastic process diffusion. Many of the advances over the original approach focused on improving the sampling speed (Chung et al., 2022; Jolicoeur-Martineau et al., 2021; Lyu et al., 2022), while others implement the noise scheduling for better modeling capacity (Nichol & Dhariwal, 2021b; Kingma et al., 2021). We can implement any of these techniques that improve general diffusion to make our methods perform faster or have better sampling quality.

Neural ordinary differential equations (Chen et al., 2018) allow capturing the irregularly sampled time series as they can naturally handle the continuous time. As such, this work is a building block which can also be used alongside our method to devise more powerful denoising networks. Rubanova et al. (2019) construct an encoder-decoder architecture based on neural ODEs which resembles the variational autoencoder (Kingma & Welling, 2014). The time series is, thus, modeled in a latent space by sampling a random vector which is propagated with an ODE. Neural stochastic differential equations (Li et al., 2020) extend this by adding noise in every solver step. This still amounts to generating the time series in a single pass, from $t_0$ to $t_{M-1}$, whereas we use the diffusion framework which refines the generated time series from pure noise $\boldsymbol{X}_N$ to the final sample $\boldsymbol{X}_0$.

Continuous-time flow process (CTFP) (Deng et al., 2020) uses normalizing flows (Kobyzev et al., 2020) to generate the time series by sampling the initial noise from the stochastic process and transform it with an invertible function to obtain the sample from the target distribution. Although this allows exact likelihood training, the method cannot capture some processes (Deng et al., 2021) and is often augmented to include a latent variable, i.e., it is trained as a VAE.

Implicit neural representations (Sitzmann et al., 2020; Dupont et al., 2022) treat data points as functions, i.e., each data point is represented by its own neural network which acts as a form of compression. In our work, we treat time series as continuous functions but instead of fitting a different neural network to each point we model the probability over the functions using the diffusion framework.

|  |  | CIR | Lorenz | OU | Predator–prey | Sine | Sink |
|---|---|---|---|---|---|---|---|
| **DSPD** | Gauss | -0.4830±0.0176 | 1.4161±0.0987 | 0.7081±0.0192 | -3.8656±0.0347 | -1.3978±0.0108 | -5.8355±0.0360 |
|  | OU | -0.4694±0.0154 | -7.3334±0.1446 | 0.5298±0.0229 | -9.4932±0.0544 | -4.2337±0.0577 | -11.3145±0.2015 |
|  | GP | -0.4512±0.0606 | **-8.4433±0.2435** | 0.5622±0.0050 | **-10.1138±0.4601** | **-4.5568±0.0732** | **-12.0081±0.0404** |
| **CSPD** | Gauss | -0.4914±0.0079 | 1.9043±0.3224 | 0.5333±0.0144 | -3.5642±0.2014 | -1.1204±0.0580 | -5.6209±0.2057 |
|  | OU | -0.4796±0.0109 | -6.6638±0.0773 | **0.4959±0.011** | -8.7748±0.1419 | -3.9496±0.1669 | -10.4739±0.2482 |
|  | GP | **-0.4935±0.0052** | -7.5565±0.4107 | 0.5251±0.0145 | -9.3409±0.199 | -4.3591±0.1242 | -11.6571±0.2825 |

Table 1: Negative log-likelihood on synthetic data (lower is better) shows OU/GP consistently better than independent noise.

|  | CIR | Lorenz | OU | Predator-prey | Sine | Sink |
|---|---|---|---|---|---|---|
| CTFP | 0.9985±0.0012 | 0.995±0.0057 | 0.783±0.0756 | 0.789±0.0227 | 0.981±0.0104 | 0.7265±0.1378 |
| Latent ODE | 1.0±0.0 | 0.998±0.0019 | 0.512±0.0331 | 0.958±0.0213 | 1.0±0.0 | 0.907±0.0394 |
| DSPD-GP (Our) | **0.5115±0.0282** | **0.5135±0.0288** | **0.5055±0.0458** | **0.5855±0.0219** | **0.5255±0.009** | **0.513±0.0103** |

Table 2: Accuracy of the discriminator trained to distinguish real data and model samples.

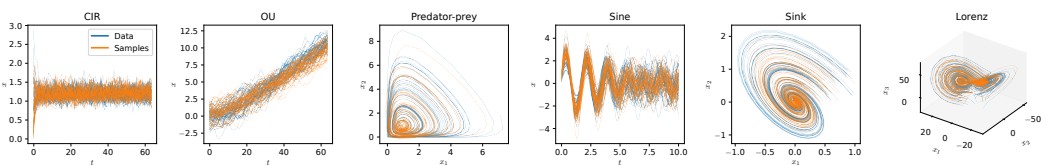

Figure 3: Real data and samples from our model based on an Ornstein-Uhlenbeck process.

## 6 EXPERIMENTS

**Probabilistic modeling.** We test our DSPD and CSPD with independent Gaussian noise and noise from stochastic processes (GP and OU) on six synthetic datasets, coming from deterministic and stochastic dynamical systems. We also compare to the established baselines for irregular time series modeling, namely, latent ODEs (Rubanova et al., 2019) and continuous-time flow process (CTFP) (Deng et al., 2020). The detailed experimental setup along with further results is in Appendix B.1.

Table 1 shows that using a stochastic process as the noise source outperforms independent noise. The ablation in Table 6 shows that using an independent denoising model, i.e., $\epsilon_\theta$ that processes time series inputs individually, performs worse than a model that processes the whole time series at once. Figure 3 demonstrates the quality of the samples. Finally, Table 2 compares our model with the baselines and demonstrates that our model produces samples that are indistinguishable to a powerful transformer-based (Vaswani et al., 2017) discriminator model. The same does not hold for the competing methods.

**Forecasting.** We test our model as defined in Section 4.1 and Figure 2 against TimeGrad (Rasul et al., 2021b) on three established real-world datasets: Electricity, Exchange and Solar (Lai et al., 2018). Due to the limitations of the CRPS-sum metric (Koochali et al., 2022), we report the NRMSE and the energy score (Gneiting & Raftery, 2007) averaged over five runs, but we note that the rank of the models' performance does not change when using other metrics as well. Table 3 shows that our method outperforms TimeGrad even though we predict over the complete forecast horizon at once, and Figure 4 demonstrates the prediction quality alongside the uncertainty estimate.

**Neural process.** We construct a dataset where each time series $X$ comes from a different stochastic process, by sampling from Gaussian processes with varying kernel parameters and time series lengths. This is a standard training setting in neural process literature (Garnelo et al., 2018). In our denoising network, we modify the attention-like layer to make it stationary (see Appendix B.2) and train as described in Section 4.2. Due to the use of tanh activations in the final layers, combined with its stationary, our model extrapolates well, i.e., when tanh saturates the mean and variance do not vary far from observations. This is the same behaviour we see in the GP with an RBF kernel, for example. The quantile loss of the unobserved data under the true GP model is 0.845 while we achieve 0.737 which indicates we capture the true process, which can also be seen in Figure 5. We remark that attentive neural process (Kim et al., 2019) does not produce the correct uncertainty.

| | TimeGrad | Ours |
|---|---|---|
| Electricity | 0.064±0.007 | **0.045±0.002** |
| | 8425±613 | **7079±164** |
| Exchange | 0.013±0.003 | **0.012±0.001** |
| | 0.057±0.002 | **0.031±0.002** |
| Solar | 0.799±0.096 | **0.757±0.026** |
| | **150±17** | 166±12 |

Table 3: NRMSE (top) and energy score on real-world forecasting data.

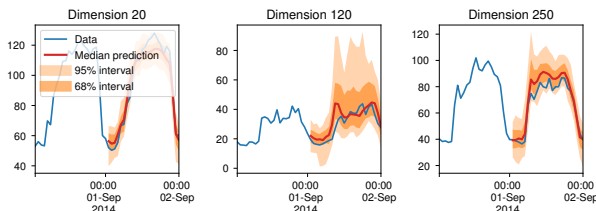

Figure 4: Forecast and uncertainty intervals on Electricity.

**Imputation.** We compare to the CSDI model (Tashiro et al., 2021) introduced in Section 4.3 on an imputation task. To this end, we use exactly the same training setup, including the random seeds and model architecture, but change the noise source to a Gaussian process. Following Tashiro et al. (2021), we use Physionet dataset (Silva et al., 2012) which is a collection of medical time series collected at hourly rate. It already contains missing values but for testing purposes we choose varying degrees of missingness and report the results on the test set. We update the loss and sampling accordingly, as in Section 3. Table 4 shows that we outperform the original CSDI model even though we only changed the noise, and the dataset we used has regular time sampling.

| Missing | CSDI | DSPD-GP (Our) |
|---|---|---|
| 10% | 0.520±0.055 | **0.498±0.036** |
| 50% | **0.644±0.024** | **0.644±0.029** |
| 90% | 0.818±0.02 | **0.815±0.019** |

Table 4: Imputation RMSE on Physionet data with varying amounts of missingness.

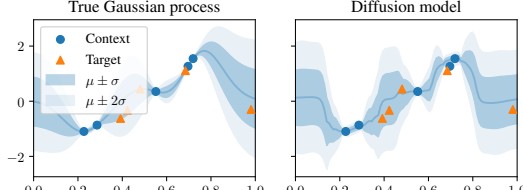

Figure 5: Sampled curves given a set of points.

# 7  DISCUSSION

In this paper, we introduced the stochastic process diffusion framework for time series modeling. We demonstrated that the improvements over the previous works come from (1) using the stochastic process as the noise source; and (2) using the model that takes in the whole time series at once, instead of modeling points independently. We also show how one can condition the generation to obtain a forecasting model as well as interpolation and imputation models.

In our experiments we used a mixture of synthetic and real-world datasets which both have regular and irregular sampling. We outperform strong baseline models on all of the tasks which demonstrates practical utility of our method.

## 7.1  FUTURE WORK

We used bare-bones diffusion without extensive tuning to demonstrate the modeling potential and make a fair comparison to other methods. However, it should be straightforward to improve upon our models by implementing recent advances in diffusion models (e.g., Nichol & Dhariwal, 2021a).

In case we have a large amount of points, we can consider replacing the current sampling strategies with more scalable variants, such as switching to a sparse GP (Quiñonero-Candela & Rasmussen, 2005). Additionally, one can train the latent diffusion (Rombach et al., 2021) by first learning the time series encoder-decoder which might be helpful for high-dimensional data, such as those we encountered in the forecasting task.

It would be interesting to explore different architecture choices, e.g., implement improvements in conditioning models via learned activations (Ramos et al., 2022). Finally, we can also apply the presented methods to other areas outside time series, such as modeling point clouds or even images, as we have demonstrated that our method is competitive on regular grids.

## ETHICS STATEMENT

We introduced a new method for time series generation. As such, it has many applications, such as probabilistic forecasting and imputation, both of which are of practical significance in the real-world settings. In particular, we would like to see successful applications in the healthcare domain. As with any generative model, one has to pay attention to the privacy and fairness when collecting data and building a model. We do not anticipate any negative outcomes applying our model on time series data. No personal information is contained in any of the datasets.

## REPRODUCIBILITY STATEMENT

Throughout the paper we describe in detail how our novel stochastic process diffusion framework works, including the algorithms for training and generation. We also describe the datasets and models we used, both in the main text and the Appendix. All the datasets we used are publicly available and the code that reproduces the results is released to the reviewers.

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

## A    DERIVATIONS

### A.1    DISCRETE DIFFUSION POSTERIOR PROBABILITY

We extend Ho et al. (2020) by using full covariance $\boldsymbol{\Sigma}(\boldsymbol{t})$ to define the noise distribution across time $\boldsymbol{t}$. If $\boldsymbol{\Sigma} = \boldsymbol{L}\boldsymbol{L}^T$ and keeping the same definitions from Section 2.1 for $\beta_n$, $\alpha_n$, and $\bar{\alpha}_n$, we can write:

$$\boldsymbol{X}_n = \sqrt{1 - \beta_n}\boldsymbol{X}_{n-1} + \sqrt{\beta_n}\boldsymbol{L}\boldsymbol{\epsilon}, \tag{17}$$

$$\boldsymbol{X}_n = \sqrt{\bar{\alpha}_n}\boldsymbol{X}_0 + \sqrt{1 - \bar{\alpha}_n}\boldsymbol{L}\boldsymbol{\epsilon}, \tag{18}$$

with $\boldsymbol{\epsilon} \in \mathcal{N}(\boldsymbol{0}, \boldsymbol{I})$. This corresponds to the following transition distributions:

$$q(\boldsymbol{X}_n|\boldsymbol{X}_{n-1}) = \mathcal{N}(\sqrt{1 - \beta_n}\boldsymbol{X}_{n-1}, \beta_n\boldsymbol{\Sigma}), \tag{19}$$

$$q(\boldsymbol{X}_n|\boldsymbol{X}_0) = \mathcal{N}(\sqrt{\bar{\alpha}_n}\boldsymbol{X}_0, (1 - \bar{\alpha}_n)\boldsymbol{\Sigma}). \tag{20}$$

We are interested in $q(\boldsymbol{X}_{n-1}|\boldsymbol{X}_n, \boldsymbol{X}_0) \propto q(\boldsymbol{X}_n|\boldsymbol{X}_{n-1})q(\boldsymbol{X}_{n-1}|\boldsymbol{X}_0)$. Since both distributions on the right-hand side are normal, the result will be normal as well. We can write the resulting distribution as $\mathcal{N}(\tilde{\boldsymbol{\mu}}, \tilde{\boldsymbol{\Sigma}})$, where:

$$\tilde{\boldsymbol{\mu}} = \boldsymbol{R}(\boldsymbol{X}_n - \boldsymbol{A}\boldsymbol{\mu}_1) + \boldsymbol{\mu}_1$$
$$\tilde{\boldsymbol{\Sigma}} = \boldsymbol{\Sigma}_1 - \boldsymbol{R}\boldsymbol{A}\boldsymbol{\Sigma}_1^T$$
$$\boldsymbol{R} = \boldsymbol{\Sigma}_1\boldsymbol{A}^T(\boldsymbol{A}\boldsymbol{\Sigma}_1\boldsymbol{A}^T + \boldsymbol{\Sigma}_2)^{-1},$$

with $\boldsymbol{A} = \sqrt{1 - \beta_n}\boldsymbol{I}$, $\boldsymbol{\mu}_1 = \sqrt{\bar{\alpha}_{n-1}}\boldsymbol{X}_0$, $\boldsymbol{\Sigma}_1 = (1 - \bar{\alpha}_{n-1})\boldsymbol{\Sigma}$, and $\boldsymbol{\Sigma}_2 = \beta_n\boldsymbol{\Sigma}$. We can now write:

$$\boldsymbol{R} = (1 - \bar{\alpha}_{n-1})\boldsymbol{\Sigma}\sqrt{1 - \beta_n}\left(\sqrt{1 - \beta_n}(1 - \bar{\alpha}_{n-1})\boldsymbol{\Sigma}\sqrt{1 - \beta_n} + \beta_n\boldsymbol{\Sigma}\right)^{-1}$$

$$= \frac{(1 - \bar{\alpha}_{n-1})\sqrt{\alpha_n}}{\alpha_n(1 - \bar{\alpha}_{n-1}) + 1 - \alpha_n}\boldsymbol{\Sigma}\boldsymbol{\Sigma}^{-1}$$

$$= \frac{1 - \bar{\alpha}_{n-1}}{1 - \bar{\alpha}_n}\sqrt{\alpha_n},$$

and from there:

$$\tilde{\boldsymbol{\mu}} = \frac{1 - \bar{\alpha}_{n-1}}{1 - \bar{\alpha}_n}\sqrt{\alpha_n}\left(\boldsymbol{X}_n - \sqrt{1 - \beta_n}\sqrt{\bar{\alpha}_{n-1}}\boldsymbol{X}_0\right) + \sqrt{\bar{\alpha}_{n-1}}\boldsymbol{X}_0$$

$$= \frac{1 - \bar{\alpha}_{n-1}}{1 - \bar{\alpha}_n}\sqrt{\alpha_n}\boldsymbol{X}_n + \sqrt{\bar{\alpha}_{n-1}}\left(1 - \frac{1 - \bar{\alpha}_{n-1}}{1 - \bar{\alpha}_n}\alpha_n\right)\boldsymbol{X}_0 \tag{21}$$

$$= \frac{1 - \bar{\alpha}_{n-1}}{1 - \bar{\alpha}_n}\sqrt{\alpha_n}\boldsymbol{X}_n + \frac{\sqrt{\bar{\alpha}_{n-1}}}{1 - \bar{\alpha}_n}\beta_n\boldsymbol{X}_0,$$

and using the fact that $\boldsymbol{\Sigma}$ is a symmetric matrix:

$$\tilde{\boldsymbol{\Sigma}} = (1 - \bar{\alpha}_{n-1})\boldsymbol{\Sigma} - \frac{1 - \bar{\alpha}_{n-1}}{1 - \bar{\alpha}_n}\sqrt{\alpha_n}\sqrt{1 - \beta_n}(1 - \bar{\alpha}_{n-1})\boldsymbol{\Sigma}^T$$

$$= \left(1 - \bar{\alpha}_{n-1} - \frac{1 - \bar{\alpha}_{n-1}}{1 - \bar{\alpha}_n}\alpha_n(1 - \bar{\alpha}_{n-1})\right)\boldsymbol{\Sigma} \tag{22}$$

$$= \frac{1 - \bar{\alpha}_{n-1}}{1 - \bar{\alpha}_n}\beta_n\boldsymbol{\Sigma}.$$

Therefore, the only difference to the derivation in Ho et al. (2020) is the $\boldsymbol{\Sigma}(\boldsymbol{t})$ instead of the identity matrix $\boldsymbol{I}$ in the covariance.

### A.2    DISCRETE DIFFUSION LOSS

We use the evidence lower bound from Equation 4. The distribution $q(\boldsymbol{X}_{n-1}|\boldsymbol{X}_n, \boldsymbol{X}_0)$ is defined as $\mathcal{N}(\tilde{\boldsymbol{\mu}}, C_1\boldsymbol{\Sigma})$, where $C_1$ is some constant (Equations 21 and 22). Similar to Ho et al. (2020),

we choose the parameterization for the reverse process $p(\boldsymbol{X}_{n-1}|\boldsymbol{X}_n) = \mathcal{N}(\boldsymbol{\mu}_\theta(\boldsymbol{X}_n, \boldsymbol{t}, n), \beta_n \boldsymbol{\Sigma})$, where:

$$\boldsymbol{\mu}_\theta(\boldsymbol{X}_n, \boldsymbol{t}, n) = \frac{1}{\sqrt{\alpha_n}} \left( \boldsymbol{X}_n - \frac{\beta_n}{\sqrt{1 - \bar{\alpha}_n}} \boldsymbol{\epsilon}_\theta(\boldsymbol{X}_n, \boldsymbol{t}, n) \right).$$

Then the KL-divergence is between two normal distributions so we can write the following, where $C_2$ is a term which does not depend on the parameters $\theta$:

$$D_{\mathrm{KL}}[q(\boldsymbol{X}_{n-1}|\boldsymbol{X}_n, \boldsymbol{X}_0)||p(\boldsymbol{X}_{n-1}|\boldsymbol{X}_n)] = D_{\mathrm{KL}}[\mathcal{N}(\tilde{\boldsymbol{\mu}}, C_1 \boldsymbol{\Sigma})||\mathcal{N}(\boldsymbol{\mu}_\theta(\boldsymbol{X}_n, \boldsymbol{t}, n), \beta_n \boldsymbol{\Sigma})]$$
$$= \frac{1}{2}(\tilde{\boldsymbol{\mu}} - \boldsymbol{\mu}_\theta)^T \boldsymbol{\Sigma}^{-1}(\tilde{\boldsymbol{\mu}} - \boldsymbol{\mu}_\theta) + C_2.$$

If we follow the implementation from Section 3.4 we reparameterize the model to predict the noise from the unit normal distribution. Then, the matrix $\boldsymbol{L}$, where $\boldsymbol{\Sigma} = \boldsymbol{L}^T \boldsymbol{L}$, can be factorized from the previous equation leaving us with the $\boldsymbol{L}^T \boldsymbol{\Sigma}^{-1} \boldsymbol{L}$ term in the middle which evaluates to an identity. We found that simplifying the loss to computing the mean squared error between the true noise $\boldsymbol{\epsilon}$ and the predicted noise $\boldsymbol{\epsilon}_\theta$, as in Ho et al. (2020), leads to faster evaluation and better results. Therefore, during training we use the loss as described in Equation 13.

Note that in the above notation we have a set of observations $\boldsymbol{X}$ for times $\boldsymbol{t}$ that we feed into the model $\boldsymbol{\epsilon}_\theta$ to predict a set of noise values $\boldsymbol{\epsilon}(t)$, $t \in \boldsymbol{t}$, whereas, previous works predicted the noise for each data point independently.

### A.3 CONTINUOUS DIFFUSION TRANSITION PROBABILITY

Given an SDE in Equation 14 we want to compute the change in the variance $\tilde{\boldsymbol{\Sigma}}_s$, where $s$ denotes the diffusion time. The derivation is similar to that in Song et al. (2021). We start with the Equation 5.51 from Särkkä & Solin (2019):

$$\frac{\mathrm{d}\tilde{\boldsymbol{\Sigma}}_s}{\mathrm{d}s} = \mathbb{E}[f(\boldsymbol{X}_s, s)(\boldsymbol{X}_s - \boldsymbol{\mu})^T] + \mathbb{E}[(\boldsymbol{X}_s - \boldsymbol{\mu})f(\boldsymbol{X}_s, s)^T] + \mathbb{E}[\boldsymbol{L}(\boldsymbol{X}_s, s)\boldsymbol{Q}\boldsymbol{L}(\boldsymbol{X}_s, s)^T],$$

where $f$ is the drift, $\boldsymbol{L}$ is the SDE diffusion term and $\boldsymbol{Q}$ is the diffusion matrix. From here, the only difference to Song et al. (2021) is in the last term; they obtain $\beta(s)\boldsymbol{I}$ while we have a full covariance matrix from the stochastic process: $\beta(s)\boldsymbol{\Sigma}$. Therefore, we only need to slightly modify the result:

$$\frac{\mathrm{d}\boldsymbol{\Sigma}_s}{\mathrm{d}s} = \beta(s)(\boldsymbol{\Sigma} - \tilde{\boldsymbol{\Sigma}}_s),$$

which will gives us the covariance of the transition probability as in Equation 15. The derivation for the mean is unchanged as our drift term is the same as in Song et al. (2021).

### A.4 SAMPLING FROM AN ORNSTEIN-UHLENBECK PROCESS

In the following, we discuss three different approaches to sampling noise $\boldsymbol{\epsilon}(\cdot)$ from an OU process defined by $\gamma$ at time points $t_0, \ldots, t_{M-1}$.

1. **Modified Wiener.** As we already mentioned in Section 3.1, we can use a time-changed and scaled Wiener process: $e^{-\gamma t} W_{e^{2\gamma t}}$. Sampling from a Wiener process is straightforward: given a set of time increments $\Delta t_0, \ldots, \Delta t_{M-1}$, we sample $M$ points independently from $\mathcal{N}(0, \Delta t_i)$ and cumulatively sum all the samples. The time changed process first needs to reparameterize the time values. The issue arises when applying the exponential for large $t$ which leads to numerical instability. This can be mitigated by re-scaling $t$.

2. **Discretized SDE.** A numerically stable approach involves *solving* the OU SDE in fixed steps. The point at $t = 0$, $\boldsymbol{\epsilon}(0)$ is sampled from unit Gaussian. After that, each point is obtained based on the previous, i.e., $i$-th point $\boldsymbol{\epsilon}(t_i)$ is calculated as $\boldsymbol{\epsilon}(t_i) = c\boldsymbol{\epsilon}(t_{i-1}) + \sqrt{1 - c^2}z$, where $c = \exp(-\gamma(t_i - t_{i-1}))$ and $z \sim \mathcal{N}(0, 1)$. This is an iterative procedure but is quite fast and stable.

3. **Multivariate normal.** Finally, we can treat the process as a multivariate normal distribution with mean zero and covariance $\mathrm{cov}(t, u) = \exp(-\gamma|t - u|)$. Given a set of time points $\boldsymbol{t}$ it is easy to obtain the covariance matrix $\boldsymbol{\Sigma}$ and its factorization $\boldsymbol{L}^T \boldsymbol{L}$. To sample, we first

| Dataset | Dim. $d$ | Dom. | Freq. | Time steps | Pred. steps |
|---|---|---|---|---|---|
| Exchange | 8 | $\mathbb{R}^+$ | day | $6,071$ | 30 |
| Solar | 137 | $\mathbb{R}^+$ | hour | $7,009$ | 24 |
| Electricity | 370 | $\mathbb{R}^+$ | hour | $5,833$ | 24 |

Table 5: Multivariate dimension, domain, frequency, total training time steps and prediction length properties of the training datasets used in the forecasting experiments.

draw $\tilde{\epsilon} \sim \mathcal{N}(\mathbf{0}, \boldsymbol{I})$ and then $\epsilon = \boldsymbol{L}\tilde{\epsilon}$. Since our model performs best if it predicts $\tilde{\epsilon}$, we opted for this particular sampling approach. If $\boldsymbol{t}$ is not changing, $\boldsymbol{L}$ can be computed once and the performance impact will be minimal. Also when sampling new realizations, $\boldsymbol{L}$ has to be computed only once, before the sampling loop (see Algorithm 2).

## B  EXPERIMENTAL DETAILS

### B.1  PROBABILISTIC MODELING

#### B.1.1  DATASETS

The properties of the open datasets used in the forecasting experiment are detailed in Table 5. Additionally, we generate 6 synthetic datasets, each with 10000 samples, that involve stochastic processes, dynamical and chaotic systems.

1. CIR (Cox-Ingersoll-Ross SDE) is the stochastic differential equations defined by:

$$\mathrm{d}x = a(b - x)\mathrm{d}t + \sigma\sqrt{x}\mathrm{d}W_t,$$

where we set $a = 1$, $b = 1.2$, $\sigma = 0.2$ and sample $x_0 \sim \mathcal{N}(0, 1)$ but only take the positive values, otherwise the $\sqrt{x}$ term is undefined. We solve for $t \in \{1, \ldots, 64\}$.

2. Lorenz is a chaotic system in three dimensions. It is governed by the following equations:

$$\dot{x} = \sigma(y - x),$$
$$\dot{y} = \rho x - y - xz,$$
$$\dot{z} = xy - \beta z,$$

where $\rho = 28$, $\sigma = 10$, $\beta = 2.667$, and $t$ is sampled 100 times, uniformly on $[0, 2]$, and $x, y, z \sim \mathcal{N}(\mathbf{0}, 100\boldsymbol{I})$.

3. Ornstein-Uhlenbeck is defined as:

$$\mathrm{d}x = (\mu t - \theta x)\mathrm{d}t + \sigma\mathrm{d}W_t,$$

with $\mu = 0.02$, $\theta = 0.1$ and $\sigma = 0.4$. We sample time the same way as for CIR.

4. Predator-prey is a 2D dynamical system defined with an ODE:

$$\dot{x} = 2/3x - 2/3xy,$$
$$\dot{y} = xy - y.$$

5. Sine dataset is generated as a mixture of 5 random sine waves $a\sin(bx + c)$, where $a \sim \mathcal{N}(3, 1)$, $b \sim \mathcal{N}(0, 0.25)$, and $c \sim \mathcal{N}(0, 1)$.

6. Sink is again a dynamical system, governed by:

$$\frac{\mathrm{d}\boldsymbol{x}}{\mathrm{d}t} = \begin{bmatrix} -4 & 10 \\ -3 & 2 \end{bmatrix}\boldsymbol{x},$$

with $\boldsymbol{x}_0 \sim \mathcal{N}(\mathbf{0}, \boldsymbol{I})$.

### B.1.2 CTFP

We implement continuous-time flow process (Deng et al., 2020) which is a normalizing flow model for stochastic processes. That is, there is a predefined base distribution $p(\boldsymbol{z})$ and a series of invertible transformations $f$ such that we can generate samples $\boldsymbol{x} = f(\boldsymbol{z})$, and evaluate the density in closed-form by computing $\boldsymbol{z} = f^{-1}(\boldsymbol{x})$ and using the change of variables formula. For more details on normalizing flows, see Kobyzev et al. (2020). The novel idea in CTFP is to change the base density to a stochastic process, i.e., a Wiener process, to obtain the distribution over the functions, similar to our work. In our case, we do not use invertible functions but learn to inverse the noising process, and additionally, we add noise at multiple levels instead only in the beginning. In the experiments we define a CTFP model as a 12-layer real NVP architecture (Dinh et al., 2017) with 2 hidden layers in each layer's MLP.

### B.1.3 LATENT ODE

Latent ODE is a variational autoencoder architecture, with an encoder that represents the complete time series as a single vector following $q(\boldsymbol{z})$, and a decoder that produces the samples at observation times $t_i$, $\boldsymbol{z}(t_i) = f(\boldsymbol{z}), \boldsymbol{z} \sim q(\boldsymbol{z})$. The final step is projection to a data space $\boldsymbol{q}(t_i) \mapsto \boldsymbol{x}(t_i)$. The key idea is to use the neural ordinary differential equation (Chen et al., 2018) to define the evolution of the latent variable $\boldsymbol{z}(\cdot)$, thus, have a probabilistic model of the function. This is different to our approach as it models the function in a latent space, with a single source of randomness at the beginning of the time series. That is, the random value is sampled at $t = 0$ and the time series is determined from there onward, whereas our method samples random values on the whole interval $[0, T]$ and does so multiple times (for $N$ diffusion steps) until we get the new realization. In the experiments we use a two layer neural network for the neural ODE, and a another two layer network for projection to the data space.

### B.1.4 OUR MODELS

We use two models, one is a simple feedforward network, and the second is an RNN-based model. The model takes in the time series $\boldsymbol{X}$, times of the observations $\boldsymbol{t}$ and the diffusion step $n$ or diffusion time $s$. The output is the same size as $\boldsymbol{X}$. The feedforward model embeds the time and the diffusion step with a positional encoding (Vaswani et al., 2017) and passes it together with $\boldsymbol{X}$ through the multilayer neural network. Here, there is no interaction between the points along the time dimension. The model, however, has the capacity to learn transformation based on time of observation. The second model is RNN based, that is, we pass the same concatenated input as before to a 2-layer bidirectional GRU (Chung et al., 2014) and use a single linear layer to project to the output dimension. Table 6 shows that it is important to have interactions in the time dimension, regardless of the noise source, because otherwise we only learn the marginal distribution and the quality of the samples suffers.

| | | CIR | Lorenz | OU | Predator-prey | Sine | Sink |
|---|---|---|---|---|---|---|---|
| **RNN-based model** | | | | | | | |
| DSPD | Gauss | 0.5245±0.0252 | 0.512±0.0212 | 0.568±0.051 | 0.5275±0.0383 | 0.5565±0.0353 | 0.526±0.0085 |
| | GP | 0.5115±0.0282 | 0.5135±0.0288 | 0.5055±0.0458 | 0.5855±0.0219 | 0.5255±0.009 | 0.513±0.0103 |
| | OU | 0.514±0.0737 | 0.6095±0.0964 | 0.5605±0.0581 | 0.5865±0.053 | 0.507±0.11 | 0.6255±0.1672 |
| CSPD | Gauss | 0.644±0.0373 | 0.5015±0.0243 | 0.6105±0.0153 | 0.548±0.0751 | 0.611±0.0516 | 0.5495±0.0313 |
| | GP | 0.5795±0.0541 | 0.674±0.0739 | 0.5025±0.0622 | 0.607±0.0538 | 0.5575±0.0376 | 0.5345±0.0201 |
| | OU | 0.4535±0.165 | 0.715±0.0884 | 0.5255±0.011 | 0.5835±0.0723 | 0.556±0.118 | 0.5795±0.0173 |
| **Feedforward model** | | | | | | | |
| DSPD | Gauss | 0.624±0.0438 | 0.713±0.1798 | 0.5275±0.0371 | 1.0±0.0 | 0.7875±0.0585 | 0.9695±0.0302 |
| | GP | 0.558±0.0611 | 0.894±0.212 | 0.5535±0.1152 | 0.7565±0.1362 | 0.735±0.2146 | 0.784±0.2281 |
| | OU | 1.0±0.0 | 1.0±0.0 | 1.0±0.0 | 1.0±0.0 | 1.0±0.0 | 1.0±0.0 |
| CSPD | Gauss | 0.537±0.0458 | 0.959±0.0808 | 0.5155±0.0165 | 0.9995±0.001 | 0.6335±0.0765 | 0.9095±0.1306 |
| | GP | 0.645±0.1034 | 1.0±0.0 | 0.507±0.0264 | 0.894±0.212 | 0.894±0.212 | 0.88±0.088 |
| | OU | 0.984±0.032 | 1.0±0.0 | 0.9905±0.019 | 1.0±0.0 | 1.0±0.0 | 1.0±0.0 |

Table 6: Accuracy of the discriminator trained on samples from a diffusion model. Values around 0.5 indicate the discriminative model cannot distinguish the model samples and real data. Values closer to 1 indicate the generative model is not capturing the data distribution.

## B.2 NEURAL PROCESS

### B.2.1 DATASET

We sample points from a Gaussian process to obtain a single time series. In the end, we have 8000 time series and 2000 test time series. We sample the number of time points from a Poisson distribution with $\lambda = 10$ but restrict the values to always be above 5 and below 50. The time points are sampled uniformly on $[0, 1]$. The observations are sampled from a multivariate normal distribution with mean zero and covariance obtained from an RBF kernel. The $\sigma$ value in the kernel is uniformly sampled in $[0.01, 0.05]$ for each time series independently. Half of the sampled points are treated as unobserved while the rest are used as a context in the model.

### B.2.2 MODEL

The denoising model takes in $\boldsymbol{X}^A$ (observed points) as a conditioning variable and $\boldsymbol{X}_n^B$ (target points) as the noisy input. We first run a learnable RBF kernel $k(\boldsymbol{t}^A, \boldsymbol{t}^B)$ to obtain a similarity matrix $\boldsymbol{K}$ between the observed and unobserved time points. We project $\boldsymbol{X}^A$ with a neural network by transforming each point independently to obtain $\boldsymbol{Z}$, and then obtain the latent variable of the same time dimension size as $\boldsymbol{X}^B$ by multiplying $\boldsymbol{K}$ and $\boldsymbol{Z}$. We then use $\boldsymbol{Z}$ as a conditioning vector and add it to projected $\boldsymbol{X}^B$, transform with a multilayer network, and obtain the output.

### B.2.3 ADDITIONAL RESULTS

We test the hypothesis that using a stochastic process with similar properties to the data will lead to better performance. The difference to the neural process setup in Section 6 is that we fix the synthetic GP to always have $\sigma = 0.05$. As can be seen from Figure 6, the marginal distribution will be equal regardless of which process and which kernel parameter we use. On the other hand, when we look at path probability $p(\boldsymbol{X})$, we notice better results when the noise process matches data properties (as was also shown in Table 1 and 6). That means, while our model can reverse the process well, the qualitative properties of the sampled curves will be different. In particular, the curves will be *rougher* with increasing $\gamma$ in OU and *smoother* with increasing $\sigma$ in GP.

## B.3 CSDI IMPUTATION

The imputation experiment presented in Sections 4.3 and 6 uses the original CSDI model (Tashiro et al., 2021) and only changes the noise to include the stochastic process source. In this case, the time points at which we evaluate the stochastic process are regular which does not reflect the true nature of the Physionet dataset. Here, we change the setup such that the measurements keep the actual time that has passed instead of rounding to the nearest hour. This is still in favour of the original paper as it only takes one measurement per hour and discards other if they are present. The model from Tashiro et al. (2021) remains the same and we replace the independent normal noise with the GP noise with $\sigma \in \{0.005, 0.01, 0.02\}$.

We run each experimental setup 10 times with different data maskings (see Tashiro et al. (2021) for more details) and report the results in Table 7. We perform the Wilcoxon one-sided signed-rank test (Conover, 1999) and reject the null hypothesis that the expected RMSE values are the same when $p < 0.05$. As we can see, higher values of $\sigma$ produce better results which makes sense since $\sigma = 0.005$ is, informally, closer to independent Gaussian sampling than $\sigma = 0.02$, which has stronger temporal dependency between the samples. We suspect 10%-missing case does not produce significant results due to noise. Using higher $\sigma$ does not further improve the results.

| Missingness: | 10% | | 50% | | 90% | |
| --- | --- | --- | --- | --- | --- | --- |
| Metrics: | RMSE | p-value | RMSE | p-value | RMSE | p-value |
| CSDI (baseline) | 0.603±0.274 | – | 0.658±0.060 | – | 0.839±0.043 | – |
| $\sigma =$   0.005 | 0.541±0.085 | 0.125 | 0.647±0.049 | 0.116 | 0.824±0.032 | 0.188 |
| 0.01 | 0.575±0.195 | 0.125 | 0.640±0.050 | **0.001** | 0.823±0.028 | **0.032** |
| 0.02 | 0.515±0.039 | 0.326 | 0.636±0.050 | **0.001** | 0.811±0.032 | **0.001** |

Table 7: Imputation results averaged over 10 runs and p-value of Wilcoxon one-sided test.

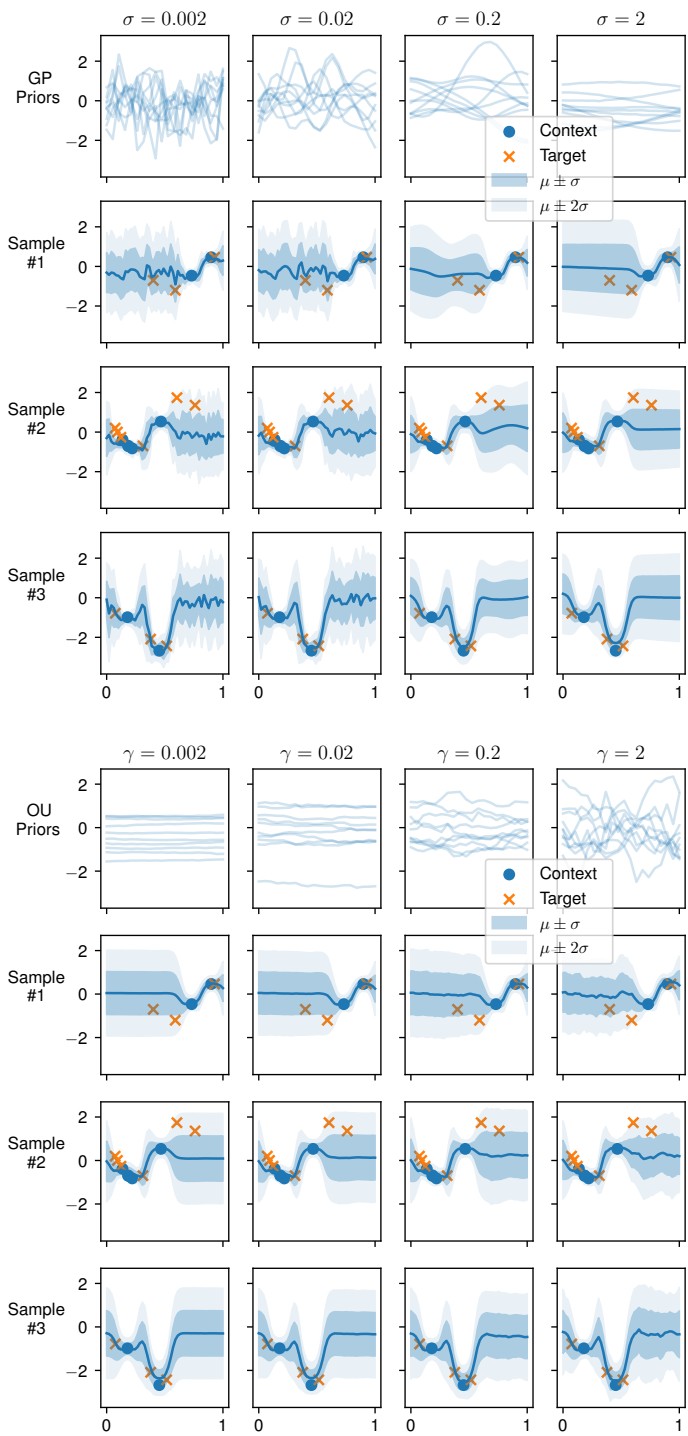

Figure 6: (Top) Neural process with Gaussian process diffusion, fitted on GP synthetic data. Columns correspond to different values of the kernel parameter $\sigma$. First row shows samples from the GP prior. As we can see, the higher the value of $\sigma$ the smoother the process is. This is also reflected in the samples from the model. (Bottom) Same but for Ornstein-Uhlenbeck process, however, increasing the kernel parameter $\gamma$ now decreases the smoothness. All of the models perfectly capture the marginal distribution.

