# OpenReview forum: "Modeling Temporal Data as Continuous Functions with Process Diffusion"
_ICLR.cc/2023/Conference — Submitted to ICLR 2023_

### Official Review · Reviewer_c2e3 · 2022-10-24

**Confidence:** 4
**Correctness:** 4
**Technical Novelty And Significance:** 2
**Empirical Novelty And Significance:** 3
**Recommendation:** 5

**Clarity, Quality, Novelty And Reproducibility:**

Question:
- On the scalability of the method, to compute the loss, the proposed method needs to compute Cholesky decompositions.The number of data points in the studied data sets seems quite big. I agree that Cholesky computaions can be done one time at the beginning of trainning. However, it can take some time to compute $\epsilon$ (line 5) in Algo. 1. How does it affect the training time?
- How to select hyperparameters ($\gamma$) for Gaussian process kernels?

Minor point:
- I’m curious about the title using the term “process diffusion”. I feel somewhat strange about this choice as I’m mostly familiar with the diffusion process as a stochastic process.


In general, the paper is fairly easy to follow. The paper mentions that the source code is available to reviewers but not found.

**Strength And Weaknesses:**

Strength:
- The paper presents an interesting idea as the covariance is introduced in diffusion processes to express some degree of correlation between temporal data points.
- For prediction tasks, the method is able to produce estimations at multiple time points unlike the previous work [Rasul et al 2021a] which follows autoregressive approaches. The key contribution here is the temporal correlation defined by covariance functions..

Weakness:
- The paper explains the approach as replacing the noise vector as a stochastic process. In fact, in a much simpler view, this way is the same with the diffusion process that just has a diffusion function as a matrix $L$ from Cholesky decomposition. In light of these observations, the contribution of formulating discrete and continuous versions of diffusion models is not significant.
- The justification for the loss function in Appendix A.2 is subtle without evidence to support the claim. In my opinion, $\Sigma^{-1}$ plays an important role in the curvature of loss functions, containing information of temporal dependencies.


**Summary Of The Paper:**

The paper proposes a diffusion model focusing on time series data where the diffusion function takes the predefined matrix from Cholesky decompositions of Gaussian processes. Experiments show some interesting results on time series predictions and imputations.


**Summary Of The Review:**

Although the paper presents a nice idea, I feel the technical contribution of the paper is limited as stochastic process for noise part is not different than matrix-valued diffusion functions for diffusion models.

---

> ### Author Response · Authors · 2022-11-13
> **Response**
>
> We would like to thank the reviewer for their positive review and constructive comments.
>
> > This approach is the same with the diffusion process that just has a diffusion function as a matrix $L$ from Cholesky decomposition.
>
> We deliberately chose the stochastic processes such that the final formulation resembles the previous works. We discuss this in great detail in Section 3.1 where we also offer an alternative in the form of a Wiener process where we would not only add matrix $L$. We argue the simplicity of our framework is one of the strengths of our approach.
>
> Formulating both the discrete and continuous versions leads to different training objectives but more importantly, we have to use different sampling algorithms. Since continuous diffusion naturally supports the use of numerical solvers (both with the SDE and ODE sampling) it might be preferred over the discrete version. Additionally, it allows for exact likelihood estimation using the probability flow.
>
> > Although the paper presents a nice idea, I feel the technical contribution of the paper is limited as stochastic process for noise part is not different than matrix-valued diffusion functions for diffusion models.
>
> We believe that stating that our work is just matrix-valued diffusion ignores the full contributions of our paper, which are defining the desirable properties of the diffusion and deriving the forward process, loss and sampling; as well as the applications in time series and generative modeling that we propose and on which we evaluate our models (that have specific architectures to satisfy certain requirements). For example, our neural process approach is a novel generative model for functions.
>
> > The justification for the loss function in Appendix A.2 is subtle without evidence to support the claim. In my opinion, $\Sigma^{-1}$ plays an important role in the curvature of loss functions, containing information of temporal dependencies.
>
> We agree that the justification in Appendix A.2 can be improved. If we follow the implementation from Section 3.4, the loss Appendix A.2 will actually include the term $L^T \Sigma^{-1} L$ where $\Sigma = L^T L$. Therefore, all the terms disappear and we get the same loss as in [1]. Note that the difference to [1] is that our model still uses different forward and reverse processes, that is, even though the loss is the same, the noisy value is obtained differently, the sampling procedure is changed and the learnable part of the model is different.
>
> > Scalability of Cholesky decomposition
>
> We do not notice any performance issues when using the Gaussian process as the noise source. It has comparable runtime as when using the independent noise since other parts of the model are more computationally demanding.
>
> As you correctly pointed out, the decomposition can also be precomputed. Finally, we propose using the Ornstein-Uhlenbeck process precisely when we might encounter performance issues, as it is possible to use more efficient sampling algorithms that are discussed in Appendix A.4. **Therefore, sampling the noise is not a computational issue in any way.**
>
> > How to select hyperparameter for Gaussian process kernels?
>
> We can treat it as a hyperparameter or learn it. Since we do not notice a big difference between the two approaches, we opt for the former. Additionally, in all of our experiments we normalize the time interval to $[0, 1]$, so using one kernel parameter value will always correspond to the same *type* of a stochastic process.
>
> We added an additional experiment using the neural process framework in Appendix B.2 which compares the effect of using a certain parameterization. As we can see from Figure 6, all the models capture the marginal distribution perfectly. However, the resulting stochastic processes have different properties. On the level of the individual path samples, the processes that have similar *smoothness* to the original data will match it better. This can also be seen from the results in Tables 1 and 6. Therefore, one can base their choice of hyperparameters using domain knowledge, performing a hyperparameter sweep, or setting them as learnable parameters.
>
> > I’m curious about the title using the term “process diffusion”. I feel somewhat strange about this choice as I’m mostly familiar with the diffusion process as a stochastic process.
>
> Our method can be best summarized as *generative diffusion models* using *stochastic processes as noise sources* for *time series/continuous functions*. We will reconsider the title to reflect this and to remove potential confusion in the camera-ready version.
>
> [1] Ho et al., DDPM, NeurIPS (2020)

---

### Official Review · Reviewer_JDSC · 2022-10-25

**Confidence:** 2
**Correctness:** 4
**Technical Novelty And Significance:** 3
**Empirical Novelty And Significance:** 3
**Recommendation:** 6

**Clarity, Quality, Novelty And Reproducibility:**

Overall the writting is good. Addressing irregular sampled data with diffusion models and modeling the time series as a continuous function is, to the best of my knowledge, new.

**Strength And Weaknesses:**

Strength:
- The abstract and introduction give a nice overview and motivation for the problem.
- Overall the paper is clear.
- The results are good and evaluation is somehow reasonable.

Weaknesses:
- I can't find the meaning of the bold format in your tables.
- You should run a statistical test to compare the different methods.
- I am not sure the editors' names are needed in the bibliography, removing them would ease the reading of it

**Summary Of The Paper:**

The authors proposes to learn a diffusion based generative model to model temporal data. The model is able to handle They test their model on various tasks such as forecasting and interpolation.

**Summary Of The Review:**

Overall I think this is a good paper tackling a well motivated and realistic task with diffusion models in the setting where temporal data are not sampled regularly.

---

> ### Author Response · Authors · 2022-11-13
> **Response**
>
> We would like to thank the reviewer for their positive review and constructive comments.
>
> > I can't find the meaning of the bold format in your tables. You should run a statistical test to compare the different methods.
>
> Bold values in tables show the best performing model, that is, the model that has the lowest average RMSE / NLL / discriminator accuracy (averaged over multiple runs). If some other model is within the standard deviation of the best average score, we make it bold as well. This convention follows previous works [e.g., 1-5]. This is often good enough since performing the statistical test or using bootstrapping for hypothesis testing would lead to the same conclusion.
>
> However, following your suggestion, we repeat the imputation experiment since this is where we see the least improvement to the alternative (due to the specific setup we use). As was already described in our paper, in this setup we do not change the model from [3] at all but rather, we change the noise, that is, we include the noise from a stochastic process with the corresponding loss. This is to see how much impact changing only the noise has on the results. We could also change the model to include continuous time in a non-trivial way and see further improvements.
>
> We compare the original CSDI [3] with independent noise and Gaussian process noise with kernel parameters 0.005, 0.01, 0.02 corresponding to increasing smoothness and temporal dependence and with varying levels of missingness (10%, 50%, 90%). We run the experiment for each of the twelve setups ([independent noise + 3xGP] x [3 missingness levels] = 12 setups) 10 times corresponding to 10 different data maskings and model initializations (see [3] for more details). After gathering the results we use the Wilcoxon test (one and two sided) to reject the hypothesis that the results have identical average values. **For further details please see Appendix B.3 in the updated pdf.**
>
> > I am not sure the editors' names are needed in the bibliography, removing them would ease the reading of it.
>
> We agree. We cleaned-up the bibliography and updated the pdf.
>
> [1] Chen et al., Neural ODE, NeurIPS (2018)\
> [2] Rubanova et al., Latent ODE, NeurIPS (2019)\
> [3] Tashiro et al., CSDI, NeurIPS (2021)\
> [4] Rasul et al., Autoregressive Denoising Diffusion, ICML (2021)\
> [5] Garnelo et al., Neural Processes, ICML workshop (2018)

---

### Official Review · Reviewer_ienA · 2022-10-25

**Confidence:** 4
**Correctness:** 4
**Technical Novelty And Significance:** 4
**Empirical Novelty And Significance:** 2
**Recommendation:** 6

**Clarity, Quality, Novelty And Reproducibility:**

The paper is quite well written, and easy to follow. The method is clearly novel, generalizing the diffusion models to time series is a nontrivial task. In one hand the proper noise process need to be used, and algorithms for interpolation, imputation and forward prediction need to be given.

Can you please give the used definition of continuity in case of stochastic process samples. What is the exact constraint you have on the noise process?

Do you have any idea on the effect of the stochastic process kernel on the model? We see the difference between OU and GP in the results. Do you have any observations you can share with the reader on the effect of the shape of the noise curve, effect of gamma parameter etc. ?

It would be useful to compare with more methods as it is not clear that the Latent ODE (Rubanova et al. 2019)  is the best alternative method. There is a concurrently published work (De Brouwer et al. 2019) that applies Bayesian filtering in the NODE framework, and a later paper (Kideger et al 2020) using the formulation of controlled differential equations and in some situation outperforms the previous two,
It is clearly a biased list given I am familiar with the NODE literature more than other similar methods.

(De Brouwer et al. 2019) De Brouwer, Edward, et al. "GRU-ODE-Bayes: Continuous modeling of sporadically-observed time series." Advances in neural information processing systems 32 (2019).
(Kideger et al 2020)  Kidger, Patrick, et al. "Neural controlled differential equations for irregular time series." Advances in Neural Information Processing Systems 33 (2020): 6696-6707.

Note that I feel the work is novel enough that it is useful for the community even if it will not over-perform every single baseline, it would still be quite useful to see a bit more benchmark, even for just a selected dataset.


**Strength And Weaknesses:**

One of the main advantage of the method is the elegant framework to combine all kinds of inference tasks for possibly sporadically observed time series. Forecasting (prediction in temporal causal direction), smoothing (prediction in both direction,) and imputation tasks can be formalized as conditional generation tasks in the framework.

One of the weakness is that little guidance is provided on how to select the noise process, and that not many benchmark method is provided, see later.

**Summary Of The Paper:**

The paper describes a novel diffusion model extending the concept to modelling irregularly sampled time series by utilizing correlated noise obtained from noise processes to produce continuous samples. The paper provide an elegant framework to combine forecasting, and imputation tasks.

**Summary Of The Review:**

The paper is quite well written and the method is novel. A little bit more detailed comparison would benefit the paper, but in general I find the paper a valuable addition to state of the art.

---

> ### Author Response · Authors · 2022-11-14
> **Response**
>
> We thank the reviewer for their positive comments and constructive feedback.
>
> > Can you please give the used definition of continuity in case of stochastic process samples. What is the exact constraint you have on the noise process?
>
> We assume data $x(t)$ is continuous in the epsilon-delta sense. Our model is almost surely continuous in a sense that $p(|x(t)-x(s)|=0) = 1$ as $t \to s$. Both Gaussian and OU processes are thus continuous.
>
> Further, we require the stochastic process to be stationary, the reasons for which we discuss in detail in Section 3.1. We can build models without this constraint (see Section 3.1) but incorporating it leads to a more convenient loss function and sampling procedure. In the neural process experiment, we also impose an exchangeability constraint.
>
> > Do you have any idea on the effect of the stochastic process kernel on the model? We see the difference between OU and GP in the results. Do you have any observations you can share with the reader on the effect of the shape of the noise curve, effect of gamma parameter etc.?
>
> We add additional results from the neural process experiment in Appendix B.2. Figure 6 shows how different kernel parameters affect the resulting samples. Increasing $\sigma$ in GP increases the smoothness of the process while increasing $\gamma$ in OU process increases roughness. However, the model is able to learn to reverse the noise in all of the cases so the marginal distribution will be essentially the same for all of the models. On the other hand, if we look at the individual sampled curves, the processes that have similar smoothness to the original data will match it better. This can also be seen from the existing results in Table 1 and 6. Therefore, one can treat the kernel parameters as a hyperparameter or make them learnable. We opted for the former approach.
>
> > It would be useful to compare with GRU-ODE-Bayes [1] and Neural CDE [2].
>
> We agree that comparing our method to additional baselines will further confirm that our approach is relevant in practice. For this reason, we perform the experiment in which we compare to [1] on the Mimic-III dataset. We use a similar setup to our forecasting experiment (Section 4.1). The dataset contains medical measurements with a high amount of missing values, which are observed at irregular times. The goal is to predict the next measurements in the fixed interval after some observation window. We use the same encoder as GRU-ODE but instead of using the *filtering* approach from [1], we learn the output distribution as in Section 4.1. That is, given the hidden state $h$ representing the history, we train a conditional diffusion process to predict the next values. Our preliminary results show that the diffusion method performs on par with the GRU-ODE model. This is not surprising since, again, the setting is similar to that in Section 4.1 and we have already shown we can model irregularly-sampled medical time series in the imputation experiment (Section 4.3). We will have full results ready for the camera-ready version.
>
> Neural CDE [2] paper, on the other hand, is orthogonal to our work as they propose a way to *represent* the irregular time series and it is not used as a generative model. Therefore, we can use it **inside our model** to encode and decode the sequence or as an encoder for the history in a forecasting experiment. We did not pursue this since the neural CDE model is not computationally efficient and other works have shown similar performance at a lower cost.
>
> [1] De Brouwer et al., GRU-ODE-Bayes, NeurIPS (2019)\
> [2] Kidger et al., Neural CDE, NeurIPS (2020)

---

### Official Review · Reviewer_8xdq · 2022-10-26

**Confidence:** 3
**Correctness:** 3
**Technical Novelty And Significance:** 3
**Empirical Novelty And Significance:** 3
**Recommendation:** 6

**Clarity, Quality, Novelty And Reproducibility:**

The paper is very well written and easy to follow. I am not sure about the level of novelty of the paper.

**Strength And Weaknesses:**

Strength: the paper is well-written and easy to follow. The proposed methodology appears to be reasonable.
Weakness: I am not so sure about the novelty of the proposed methodology.

1. Although the authors repeatedly emphasized about the denoising approach is designed for stochastic processes, the loss function (13) and (8) with score function (16) are still vector-based loss functions. It would make more sense to generate the noise process from a stochastic process instead of a multivariate normal distribution. Even if one needs to discretize the stochastic process, the dimension of the noise process should be higher than the original observed data. Under the current setting, it is really difficult to tell how much difference is there between the proposed method and those in[1] and [2], from the implementation point of view.

2. When considering the stochastic process, it is not as simple as "Gaussian+Gaussian" is still Gaussian. Even Gaussian processes with different covariance kernels have different properties such as smoothness and differentiability. This brings another question: what noise processes to use in practice? The paper proposes two candidates: a Gaussian process with a radial basis kernel and the Ornstein-Uhlenbeck process. Any guidelines on how to choose? Do parameters in the kernel function need to be estimated? Is it possible that a noise process with similar smoothness/differentiability/strength-of-dependence properties to the original data will yield better performance? These issues need to be investigated.


Reference:

[1] Ho, Jonathan, Ajay Jain, and Pieter Abbeel. "Denoising diffusion probabilistic models." Advances in Neural Information Processing Systems 33 (2020): 6840-6851.

[2] Song, Yang, Jascha Sohl-Dickstein, Diederik P. Kingma, Abhishek Kumar, Stefano Ermon, and Ben Poole. "Score-Based Generative Modeling through Stochastic Differential Equations." In International Conference on Learning Representations. 2020.

**Summary Of The Paper:**

This paper proposes to model time series data as discretized observations from an underlying continuous stochastic process using a diffusion framework. The main contribution of the paper is to extend the method proposed in [1] and [2] to the case where a stochastic noise process is used for the forward process. Based on the proposed method, the paper also developed approaches for multivariate probabilistic forecasting and imputation tasks.  The effectiveness of the proposed method is demonstrated through synthetic and real-world datasets.


Reference:

[1] Ho, Jonathan, Ajay Jain, and Pieter Abbeel. "Denoising diffusion probabilistic models." Advances in Neural Information Processing Systems 33 (2020): 6840-6851.

[2] Song, Yang, Jascha Sohl-Dickstein, Diederik P. Kingma, Abhishek Kumar, Stefano Ermon, and Ben Poole. "Score-Based Generative Modeling through Stochastic Differential Equations." In International Conference on Learning Representations. 2020.

**Summary Of The Review:**

The paper is very well written and easy to follow. I am not sure about the level of novelty of the paper.

---

> ### Author Response · Authors · 2022-11-13
> **Response**
>
> We would like to thank the reviewer for their positive review and constructive comments.
>
> > The loss function (13) and (8) with score function (16) are still vector-based loss functions.
>
> The loss function is in fact minimizing the negative log-likelihood $-p(X)$ where $X$ is the observed time series which is **a set of vectors**, whose length is not fixed. Note that both the noise and the model’s output are not independent vectors, as was the case in the previous works (e.g., [1] and [2]), but are instead correlated noise from a stochastic process. Therefore, we are always operating on the whole time series at once and the loss reflects that.
>
> > It would make more sense to generate the noise process from a stochastic process instead of a multivariate normal distribution.
>
> This is exactly what we do. The Gaussian process and the Ornstein-Uhlenbeck are both stochastic processes [3,4]. Additionally, our architecture in the neural process experiments (Section 4.3, 6 and Appendix B.2) satisfies the exchangeability condition so the whole model generates samples from a stochastic process. Our neural process model is also independent of the number of time points t at which we sample realizations x(t), so we can successfully generate continuous functions, which we demonstrate in the experiments (see also Figure 6 in Appendix B.2).
>
> > Even if one needs to discretize the stochastic process, the dimension of the noise process should be higher than the original observed data.
>
> The stochastic process can be thought of as infinite-dimensional since it is generating functions, and we sample a finite number of points by evaluating the generated function using time t as an input argument. This fact is what allows us to do interpolation in a principled way, as we show in the neural process experiment. Without using stochastic process noise, our model would generate discontinuous outputs.
>
> > Implementation difference between the proposed method and those in [1] and [2]?
>
> The main implementation differences are: 1) the noise is not independent in our model so we use different forward noising process and different objective; 2) the model is inputting and outputting the whole time series as an encoder-decoder to match the noise, that is, we need different architectures to [1] and [2]; 3) depending on the application (forecasting, imputation, etc.) we have additional components in the model that enable, e.g., conditioning on the history.
>
> We want to re-emphasize that the contributions of our paper are establishing the framework that works on continuous functions while keeping the nice properties that appear in the previous works (e.g., [1] and [2]). Consider, as an alternative, the Wiener process as the noise source (which we already discuss in Section 3.1). This would lead to a different loss and different properties which might not be desirable (see Section 3.1). Thus, our choice of noise sources and the resulting losses is **deliberate** to satisfy the imposed constraints which at the same time leads to simpler implementation for the practitioners.
>
> > Any guidelines on how to choose between GP and OU process?
>
> From the empirical results there are no significant differences between the two choices. However, the Gaussian process can be informally viewed as generating *smoother* functions. At the same time, sampling from an Ornstein-Uhlenbeck scales *nicely* with the number of points so it is more suitable when we have a very large number of points (however, we did not encounter these issues with our datasets).
>
> The parameters in the kernel function can be selected beforehand as a hyperparameter but can be learned as well. We do not notice a significant difference between the two approaches and we use the former approach in the paper.
>
> We test the hypothesis that using a stochastic process with similar properties to the data will lead to better performance **in Appendix B.2**. As can be seen from Figure 6, the *marginal distribution* will be equal regardless of which process and which kernel parameter we use. On the other hand, when we look at path probability $p(X)$, we notice better results when the noise process matches data properties (see Table 1 and 6). That means, while our model can reverse the process well, the qualitative properties of the sampled curves will be different. In particular, they will be *rougher* with increasing $\gamma$ in OU and *smoother* with increasing $\sigma$ in GP. Note that if we do not use a stochastic process as noise source but rely on independent normal distribution, the resulting samples will not be continuous and we empirically show the results will be worse.
>
> [1] Ho et al., DDPM, NeurIPS (2020)\
> [2] Song et al., Score-based generative modeling through stochastic differential equations, ICLR (2021)\
> [3] Särkkä, Simo, and Arno Solin. Applied stochastic differential equations. Cambridge University Press (2019)\
> [4] Williams & Rasmussen. Gaussian processes for machine learning. MIT press (2006)

---

### Author Response · Authors · 2022-12-07
**Gentle reminder**

We would like to thank all the reviewers for their constructive feedback. Since the review period is ending soon we would like to encourage the reviewers to join the discussion. We hope that our response and updated manuscript cleared the misconceptions and addressed all the questions. If there are any further questions we will happily answer them.

---

### Decision · Program_Chairs · 2023-01-20

**Decision:**

Reject

**Justification For Why Not Higher Score:**

In sum, the paper provides an interesting and promising model for temporal data. However, there are still several points not clarified. I encourage the authors to consider the reviewers' suggestion to improve the paper.

**Justification For Why Not Lower Score:**

N/A

**Metareview: Summary, Strengths And Weaknesses:**


In this paper, the authors extend the diffusion model for modeling functional data, which are represented as a set of points with different length.

The paper is well-written and easy-to-follow.

The major concerns raised by the reviewers:

- Although the paper is trying to establish the functional generalization of diffusion model, the current presentation does not emphasize this, which makes the reviewers confused.

- The experiment part is relatively weak. As the reviewer mentioned, the current competitors are limited. More competitors, e.g., GRU-ODE and other neural processes, should be included to demonstrate the benefits.


Minors:

- The current derivation eventually leads to Gaussian processes. A more detailed discussion might be required to show the benefits over the classic Gaussian processes, especially with parametrized kernel learning.

- The limited distribution still follows Gaussian, according to Eq. 15. Does this mean the model can only be applied to regression? How this model can be used for classification?